# A Comprehensive Analysis of Inorganic Ions and Their Selective Removal from the Reconstituted Tobacco Extract Using Electrodialysis

**DOI:** 10.3390/membranes12060597

**Published:** 2022-06-07

**Authors:** Shaolin Ge, Qian Chen, Zhao Zhang, Shike She, Bingxia Xu, Fei Liu, Noor Ul Afsar

**Affiliations:** 1IAT USTC-AHZY Joint Laboratory of Chemistry & Combustion, Institute of Advanced Technology, University of Science and Technology of China, Hefei 230088, China; slge@mail.ustc.edu.cn; 2Anhui Key Laboratory of Tobacco Chemistry, Anhui Tobacco Industrial Co., Ltd., 9 Tianda Road, Hefei 230088, China; zhaoz@mail.ustc.edu.cn (Z.Z.); m15638147593@163.com (B.X.); feiliu@ahycgy.com.cn (F.L.); 3Applied Engineering Technology Research Center for Functional Membranes, Institute of Advanced Technology, University of Science and Technology of China, Hefei 230088, China; chenaqian@mail.ustc.edu.cn; 4Key Laboratory of Combustion & Pyrolysis Study of CNTC, Anhui Tobacco Industrial Co., Ltd., 9 Tianda Road, Hefei 230088, China; sheshike@hotmail.com; 5Anhui Provincial Engineering Laboratory of Functional Membrane Materials and Technology, Department of Applied Chemistry, School of Chemistry and Materials Science, University of Science and Technology of China, Hefei 230026, China

**Keywords:** reconstituted tobacco extract, nicotine, electrodialysis, ion exchange membranes

## Abstract

Many tobacco stalks, dust, and fines are discharged in the tobacco industry, rich in inorganic minerals ions and nicotine salts. The high salinity and nicotine salts are challenging to be addressed by traditional treatment and are a severe threat that ought to be overcome. Thus, proper techniques can regenerate the tobacco stalks into reconstituted tobacco flakes used as cigarette filler. The electrodialysis process has been a viable approach to removing the inorganic ingredients in wastewater. We studied concentration, pH, and co-related influences with the nicotine and sugar/nicotine contents on the desalination performance. The results show that the inorganic ions such as Cl^−^, K^+^, Ca^2+^, and Mg^2+^ ions were successfully removed. When the feed concentration ranges from 3 to 15%, the removal ratio of the K^+^ ions is higher than Ca^2+^ and Mg^2+^ ions. As we reported previously, the K^+^ and Ca^2+^ ions are unfavorable for the total particulate matter emission but beneficial to decreasing the HCN delivery in mainstream cigarette smoke. Selective ED is a robust technology to reduce the harmful component delivery in cigarette smoke.

## 1. Introduction

Many tobacco stems, dust, and fines are produced as waste in the tobacco industry, nearly one-third of the raw materials [1]. The papermaking process reconstitutes tobacco product that contains tobacco stems and fines as primary raw materials and is produced using papermaking technology. The principal procedure is as follows: (i) place tobacco raw material into the water to isolate soluble and insoluble; (ii) the insoluble will depart via the refining process and paper machine for fiber sheet while the soluble extracts will be sized back to the base sheet after being concentrated. The sized sheet will be dried and ultimately assembled into PM recon. Analogized to natural tobacco, tobacco produced from reconstituted tobacco has several advantages in structural stability, combustion performance, and tar delivery [2]. The low chemical concentration and rapid combustion of PRT when compared to natural tobacco can reduce the inhaled nicotine and lower the puff number of cigarettes [3]. It is reported that the PRT comprises 20–25% of the tobacco raw materials for several high-grade cigarettes such as Marlboro, Kent, Winston, and Camel [4]. In contrast to natural tobacco leaf, papermaking tobacco has inorganic constituents such as K^+^, Cl^−^, SO_4_^2−^, and NO_3_^−^. Multiple investigations have revealed that the ionic elements significantly impact the sensory taste of the cigars and the thermal behavior in cigarette smoke [5]. Generally, Cl^−^ and NO_3_^−^ ions are considered undesirable components that affect cigarettes’ moisture absorption and combustion value and harmful carbonyl compounds delivery in mainstream smoke [6,7]. In contrast, divalent or multivalent alkaline metal ions are usually considered desirable components that have catalytic roles in the thermal degradation and char formation of biomass [8]. Therefore, it is necessary to control the ionic components of the tobacco extract manually [9,10].

For the removal of undesired inorganic matter, several separation technologies such as multistage continuous extraction [11], solvent extraction, condensation, ion exchange [12], and absorption were applied [13]. Generally, these techniques revealed unsatisfying performances on selective removal of inorganic ions and have drawbacks such as extensive consumption of chemicals, high operating cost, and being the specific cause of secondary pollution. Contrary to the separation technologies, electrodialysis (ED) does not suffer from major flaws such as costly chemicals, production of large amounts of waste, and short lifetimes of absorbents. Therefore, ED has been deemed an environmentally acceptable technology, which has found many applications in water desalination, cleaning production or separation, resources recycling, and power generation [14,15,16,17,18]. Bazinet and co-authors [19] used the ED process for the electromigration of polyphenols in tobacco extract, with an overall demineralization of 77%. In another study [20], the authors further enhanced the separation performances of polyphenols from tobacco by extending the experimental time. Similarly, our previous studies have proven the feasibility of ED for withdrawing inorganic ions and lowering the harmful ingredient in cigarette smoke [21,22,23].

In these studies, ED was used to dispose of Cl^−^ and NO_3_^−^ ions and selectively revise the monovalent and divalent inorganic ion composition. During the combustion of cigarettes, the primary toxic volatile compounds such as CO, NH_3_, HCN, phenol, and crotonaldehyde are discharged from the pyrolysis of carbohydrates. Our earlier studies [22,23] have demonstrated that ED helps reduce the harmful components of cigarette smoke. The reconstituted tobacco extract with ED treatment showed improved performance on the taste and delivery of toxic mainstream tobacco smoke compared to that without the ED treatment. However, in the industrial application of ED on tobacco sheet extract, the stability of membranes is the foremost important thing to be considered [24,25,26]. The compositions of tobacco extract are very complicated, including thousands of kinds of matter such as minerals, sugars, organic acids, amino acids, carbohydrates, polyphenols, esters, alcohols, pigments, etc. [27,28,29]. However, infrequent analyses of the reconstituted tobacco extract characteristics, such as the concentration, pH, current density, voltage variation, sugar/nicotine, and sugar/nicotine for treating tobacco sheet extract from a practical production line, have never been analyzed in detail. Herein, we developed readily adaptable methods to monitor the desalination of reconstituted tobacco extract and elucidated the evolution of the ED process based on ion-selective membranes. Therefore, the main objectives of this research were to investigate the stability of the ED process for desalting reconstituted tobacco extract and shed light on the application of ED for industrial application.

## 2. Experimental Section

### 2.1. Chemical and Materials

Reconstituted tobacco extract of different concentrations used in this investigation was supplied by China Tobacco Anhui Industrial CO., Ltd., Hefei, Anhui Province, China. The tobacco extract was centrifuged at 3 k rpm for five min to clear the undissolved solids. Two ion exchange membranes, i.e., CJ-MA-2 and CJ-MC-2 (supplied by Hefei ChemJoy Polymers Co., Ltd., Hefei, Anhui, China), and their detailed properties are given in Table 1. All the chemicals used in this work were of analytical grade.

### 2.2. Electrodialysis Experiment

An elaborate schematic diagram of the ED setup is shown in Figure 1. The ED setup comprised: (1) a cathode and an anode clinched on organic glass plates on both sides; the electrodes were assembled from iridium-tantalum with a thickness of 1.5 mm. A DC power supply (WYL1703, Hangzhou Siling Electrical Instrument Ltd.) was attached to the two electrodes. The voltage drop across the stacks was directly recorded from the power supply screen. (2) We set eleven pieces of cation exchange membranes (CEMs) and ten pieces of anion exchange membranes (AEMs) with an effective area of each membrane (189 cm^2^) alternatively. (3) Sealing spacers of PE with a thickness of 0.75 mm were fixed to separate the membranes; (4) and beakers to maintain the feed volume of 500 mL. Each beaker was linked with a submersible pump (AP1000, Zhongshan Zhenghua Electronics Co., Ltd., China, with a flow rate of 22 L h^−1^). Our experimentations assigned an electrode, concentrate, and diluted chamber, established in this ED stack. A 400 mL Na_2_SO_4_ solution (0.3 mol/L) was supplied to the electrode chambers as an electrolyte solution. A total of 400 mL of reconstituted tobacco extract and tap water was fed into the diluted and concentrate chambers, respectively. Each chamber was bathed for 30 min before the experiment to eliminate visible bubbles.

### 2.3. Analytical Methods

We measured the conductivity of the diluted chamber by a conductivity meter (DDS 307, Shanghai INESA Scientific Instrument Co., Ltd., Shanghai, China). The concentration of inorganic ions was determined by ion chromatography (ICS3000 multifunctional ion chromatography and ED electrochemical detector, DIONEX Company, Sunnyvale, CA, USA). HPLC was used to measure the concentration of sugar and nicotine (Acquity H-Class UPLC, Waters Corp., Milford, MA, USA; 4000Q Trap, AB Sciex, Redwood City, CA, USA).

### 2.4. Physiochemical Properties of IEMs

The water uptake (WU) was measured by saturating the membranes with water for one day and denoting them as $W_{2}$. The wet membranes were dried at 60 °C for 6 h, and reweighed as $W_{1}$. The WU of the membranes was calculated using the following Equation (1):(1)$$\mathrm{WU}\;=\frac{\left(W_{2}-{\;W}_{1}\right)}{W_{1}}\times 100$$

W_1_ (g) and W_2_ (g) represent weight of the dried and wet membrane, respectively.

The membranes’ anion exchange capacity (AEC) and cation exchange capacity (CEC) were measured by immersing the membranes in a Na_2_SO_4_ (0.5 mol L^−1^) solution for 24 h to convert the membrane into the SO_4_^2−^ form. Finally, the amount of Cl^−^ ions was titrated with a AgNO_3_ (0.1 mol L^−1^) aqueous solution using K_2_CrO_4_ as an indicator. The AEC values were determined by the amount of AgNO_3_ consumed in the titration and the mass of the dry membrane (Equation (2)). Similarly, the CEC of CEM (H^+^ form) was measured by titrating the amount of H^+^ ions released with the standardized NaOH solution (0.1 mol L^−1^) using phenolphthalein as an acid-base indicator (Equation (3)).
(2)$$\mathrm{AEC}=\frac{C_{{\mathrm{AgNO}}_{3}}V_{{\mathrm{AgNO}}_{3}}}{W_{\mathrm{Dry}}}$$
(3)$$\mathrm{CEC}=\frac{C_{NaOH}V_{NaOH}}{W_{\mathrm{Dry}}}$$

The transport number of the counter ion was measured by using potential difference across the membrane. Next, 0.05 mol L^−1^ NaCl and 0.01 mol L^−1^ NaCl solutions were used in a two-compartment cell. The potential difference between the two reference electrodes was measured. The transport number was calculated by Equation (4) [30];
(4)$$E_{m}=\frac{\mathrm{RT}}{F}\left(2\bar{t_{+}}-1\right)\ln\frac{C_{1}}{C_{2}}$$
where E_m_ is the electrode potential difference; R the universal gas constant; F the Faraday’s constant; T is the absolute temperature; t_+_ is the transport number; C_1_ and C_2_ are the molar concentrations in both dilute and concentrated cell chambers, respectively.

The membrane area resistance was measured using a custom-made, four-compartment, module-lab-made equipment. The membranes under examination were placed in the center compartment, and two identical commercial CEMs were used on either side of the electrolyte. Here, 0.3 mol L^−1^ Na_2_SO_4_ solution was used to rinse the electrodes and 0.5 mol L^−1^ NaCl solution was supplied to the intermediate chamber [31]. The voltage drop across an IEM was measured with a multimeter attached with a pair of Ag-AgCl electrodes placed near the surface of the membrane. The membrane area resistance (R_M_) was given by [32];
(5)$$R_{m}=\frac{V-V_{0}}{I}A$$

In Equation (5), R_m_ is the membrane’s area resistance (Ω·cm^2^); V and V_0_ are potentials (volts) with and without the membrane; I is the current and A is the effective area of the membrane.

## 3. Results and Discussion

In this ED experiment, to avoid concentration polarization, the membrane’s voltage was kept below 10 V. The composition of reconstituted tobacco liquid is complicated and challenging, so its concentration has a remarkable impact on the ED process; the higher the concentration, the higher the viscosity of the reconstituted tobacco extract; it has poor fluidity and influences the pretreatment process and desalination [22]. So, we examined the consequence of different feed concentrations on the ED process. Figure 2 illustrates the current and voltage changes concerning feed concentration for 3, 5, 7, 10, and 15 wt% tobacco extract in the ED stack. As shown in Figure 2, at different concentrations of tobacco liquid, the voltage of the membrane stake slowly increases at the beginning phase. When it arrives at the maximum value, it begins to maintain a constant voltage. The current of the membrane stake declines gradually over time. From the variation of voltage and current with time, it can be noticed that the resistance of the membrane stakes increases gradually with the ED process. This shows that the ion concentration in the diluted chamber is decreasing. The pH of different concentrations of tobacco extract varied during the ED process, whereas the pH value decreased with the increased concentration of tobacco refined liquor. It may be because with the increase of concentration of the tobacco extract solution, the concentration of inorganic salt ions in the solution also increases, and the concentration polarization reaction under the same membrane pair voltage decreases gradually. Therefore, the concentration polarization phenomenon can be reduced by increasing the concentration of refined liquor of tobacco. At the same time, the concentration of tobacco refined liquor greatly influences ion migration. When the concentration increases, the viscosity of feed liquid increases, and the ion migration rate slows down. Therefore, the desalination efficiency of tobacco liquid with low concentration is higher. The conductivities of tobacco extract with lower concentrations are slightly lower and require less time to desalinate than the higher concentration. It is rational since the desalination rate is proportionate to the initial concentration. The experiments were halted when the conductivities in the feed solution were not decreased anymore.

### 3.1. The Nicotine Content in Different Tobacco Liquids

During the ED process, changes in nicotine content and total sugar/nicotine in tobacco liquid were also examined. As shown in Figure 3, the nicotine loss rate increased with time. The ratio of total sugar to nicotine ranged from 10 to 65. The sugar/nicotine ratio trend is consistent with total sugar. The loss of nicotine decreases when increasing the concentration. According to these characteristics, ED can control tobacco liquid’s sugar/nicotine ratio and improve the quality.

### 3.2. Ions Concentration of Tobacco Extract

According to ion chromatography, the change of ion concentration in the desalination and concentrated chambers of tobacco liquor with different concentrations is shown in Figure 4. It has been noted that the removal rates of K^+^ and Cl^−^ of the feed reached the maximum and increased in the concentration compartment with the changes in concentration. Similarly, the removal rates of divalent cations (Mg^2+^ and Ca^2+^) showed lower reduction due to the membrane selectivity for these divalent ions. Usually, the removal ratio of monovalent ions is higher than that of the divalent ions due to having smaller hydration radii and lower charge density. We can conclude from the ED results that the inorganic salts in tobacco liquor can be removed and collected in the concentrated chamber via the ED process.

### 3.3. ED Analysis at Different pH Values

According to the results of the ED of different concentrations of feed liquid, the pH has a great influence on the removal of inorganic salts and nicotine. In Figure 5, at different pH, the voltage of the membrane stakes gradually increases to the maximum value, and the constant voltage begins to be active. The current of the membrane stakes rises first with time and gradually decreases after reaching the ultimate value, which is due to the start of NaOH or HCl in adjusting the pH value so that the initial conductivity rises. At the same time, the change range of pH was minute during the whole ED process, and the ion removal was reduced compared with the tobacco liquid without adjusting the pH value. Nicotine can be changed into an ionic state under the influence of pH value. By adjusting the pH value, nicotine can be removed from the concentrated chamber so that inorganic salts and nicotine can be simultaneously augmented in the concentrated chamber.

### 3.4. Total Sugar/Nicotine and Reducing Sugar/Nicotine at Different pH Values

At the same time, we also investigated the changes in nicotine content and total sugar/nicotine in tobacco liquid under different pH values, as shown in Figure 6. It was found that nicotine loss rate was directly linked to pH value. The ratio of total sugar to nicotine decreased with increasing the pH value. It indicates that pH greatly impacts the molecular state of nicotine during the ED analysis. We can prevent the loss of nicotine by adjusting the pH value. The ratio of sugar to nicotine would give a balance of opposing effects and thus serve as a good smoking quality indicator. A high percentage may indicate mildness and smoothness, while a meagre ratio may indicate harsh, irritating smoke. If the ratio is too high, it may mean that the tobacco is too mild to be pleasing to the smoker. High sugar content agreeing with nicotine level is the most desirable quality for smoking.

### 3.5. The Influence of pH on the Ion’s Removal

In energy-efficient ED desalination procedures, salt ions are transported through IEMs from high salinity to low salinity. Besides salt ions, organic micropollutants can also be transported from wastewater containing high organics to low organics. The passage mechanisms of organic micropollutants via IEMs are complicated phenomena, and pH influences them enormously. Since pH variations in the ED are common, it is crucial to investigate their impact on these mechanisms. From Figure 7, it can be observed that the concentration was higher for K^+^ and Cl^−^, which revealed high transport for monovalent ions, while for divalent cations such as Ca^2+^ and Mg^2+^, the effect was minor. Similarly, the ED can also cause pH changes during desalination, resulting in salt precipitation of Mg^2+^ and Ca^2+^ and consequently affecting the process’s desalination performance and long-term stability. The solution pH affects the ED separation of ions in multiple ways: (a) solution pH will decide the ionic species’ charge that influences the rate and permselectivity of ion migration across the IEM; (b) a higher pH may result in the precipitation of divalent ions, lowering their availability; (c) at very low pH, the excess H^+^ can carry the charges across the membrane, impeding the transport of other ions; (d) acidic pH is advantageous for the leaching of inorganic ions in pretreatment techniques.

## 4. Conclusions

The laboratory-scale experimental setup was used to desalt tobacco extract using the ED process. It was observed that high currents could accelerate the desalting efficiency but would induce higher energy consumption, whereas low currents have high current utilization but require a higher capital cost. For tobacco extract of different concentrations, monovalent ions’ removal ratios are as high as for divalent ions. A low tobacco extract is suitable content by considering the desalination and treatment capacity. The ED technology is a potential technology to enhance the smoking quality of cigarettes and selectively reduce the harmful component in cigarette smoke. However, it needs an interdisciplinary partnership between academia and industry from the membrane science and tobacco industry to identify the detailed connection between the inorganic ingredients and cigarette smoke delivery.

## Figures and Tables

**Figure 1 membranes-12-00597-f001:**
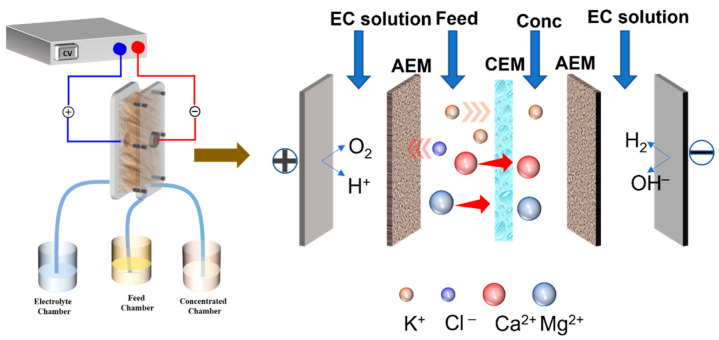
Schematic diagram for the experimental stack and the main principle of the ED process.

**Figure 2 membranes-12-00597-f002:**
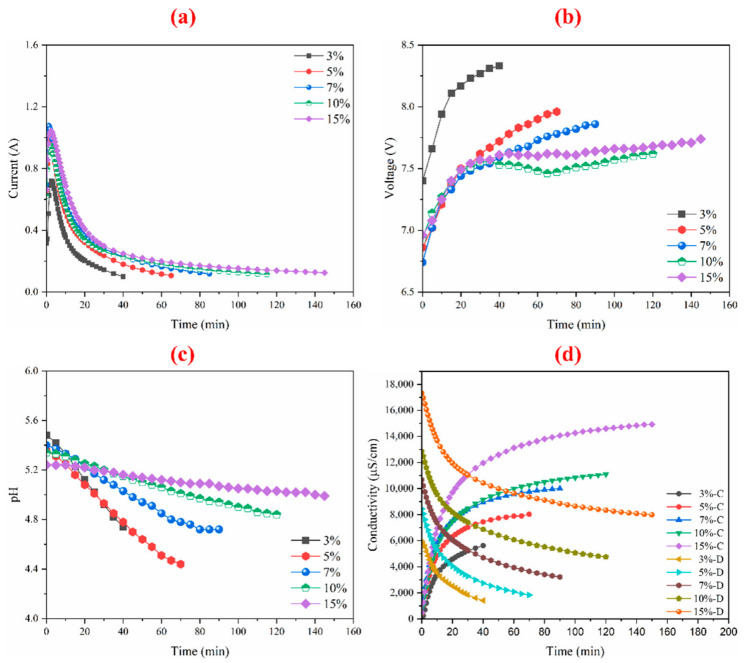
Concentration influence on current density (**a**), voltage drop (**b**), pH (**c**), and conductivity in ED process (**d**).

**Figure 3 membranes-12-00597-f003:**
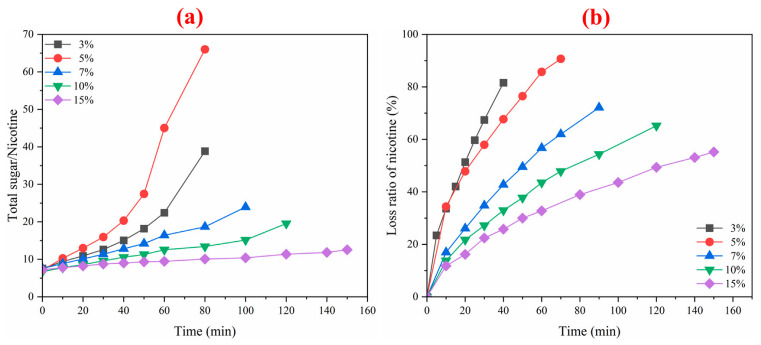
Variation curves of total sugar/nicotine (**a**) and loss ratio of nicotine (%) (**b**) in different concentrations of tobacco liquids.

**Figure 4 membranes-12-00597-f004:**
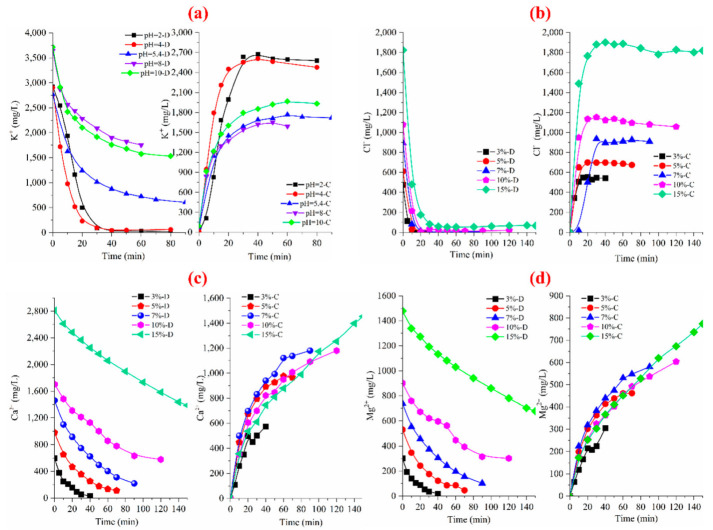
The alteration of ion contents: (**a**) K^+^, (**b**) Cl^−^, (**c**) Ca^2+^, (**d**) Mg^2+^ ions in diluted and concentrated chambers of different concentrations of the tobacco extract.

**Figure 5 membranes-12-00597-f005:**
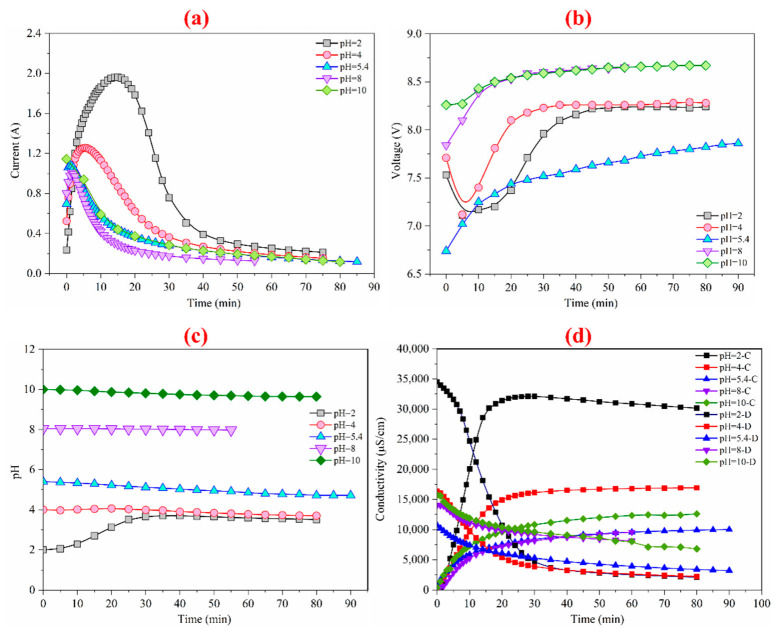
Variation in the current (**a**), voltage drop (**b**), pH (**c**), and conductivity (**d**) at different pH values.

**Figure 6 membranes-12-00597-f006:**
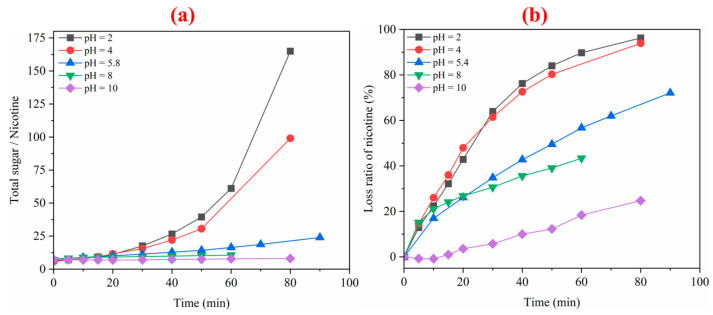
The effect of pH alteration on the total sugar/nicotine content (**a**) and loss ratio of nicotine (**b**) in the ED process.

**Figure 7 membranes-12-00597-f007:**
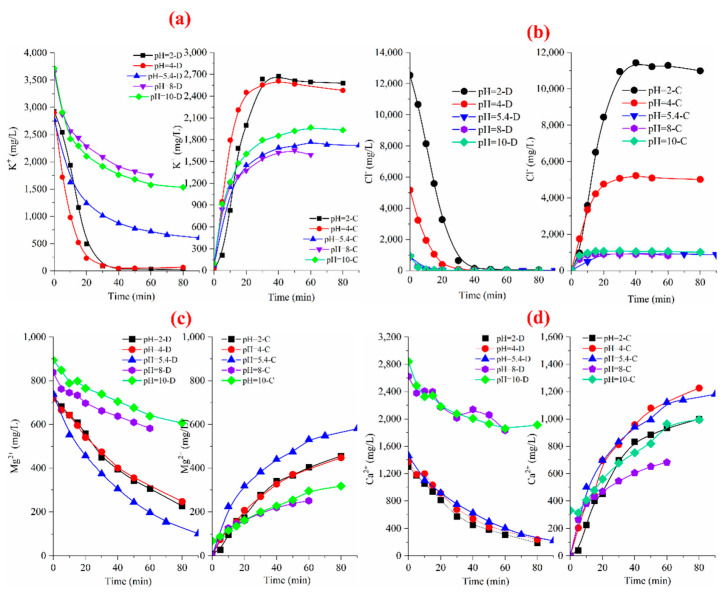
Variation curves of (**a**) K^+^, (**b**) Cl^−^, (**c**) Ca^2+^, (**d**) Mg^2+^ ions in the concentration and desalination chambers at different pH values.

**Table 1 membranes-12-00597-t001:** The characteristics of the membranes used in the experiments.

Membranes	Thickness (mm)	IECs (mmol g^−1^)	WU (%)	R_M_ (Ω·cm^2^)	Transfer Number	Break Stress (MPa)
CJ-MC-2	0.200	1.50	35	1.5	0.98	>3.5
CJ-MA-2	0.145	1.25	32	1.2	0.99	>3.5

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
