# Peer review of "A Comprehensive Analysis of Inorganic Ions and Their Selective Removal from the Reconstituted Tobacco Extract Using Electrodialysis"

_membranes, 2022, doi:10.3390/membranes12060597_

Round 1

Reviewer 1 Report

The paper 'A comprehensive analysis of inorganic ions and their selective removal from the reconstituted tobacco extract using an electrodialysis' by Noor et al. summarizes the reconstituted tobacco extract findings. However, it is not just a 'paper about papers' but delivers further analysis and endeavors to extract the most noteworthy results. It is well written and well balanced between industrial and academic collaboration. It deserves to be published and is a practical contribution to this journal. In order to make it more readily accessible, several questions and comments were raised when reading the article:

1.       It is suggested to redraft the abstract section and make it more conclusive.

2.       The figures' quality is inferior, and it is hard for the reader to extract a finding.

3.       Please provide more detail about the title of each figure, and cite it in the main text.

4.       Section 3.1 and 3.4 needs to be combined/discussed in one section.

5.       How has the concentration of ions been calculated? Please provide a mathematical equation.

6.       Please add some recent examples and references for ion-selective ion exchange membranes and ED applications, such as Journal of Membrane Science, 2021, 634, 119407; Desalination 2022, 531,115690, etc.

Author Response

Reviewer 1

In the article "A comprehensive analysis of inorganic ions and their selective removal from the reconstituted tobacco extract using an electrodialysis" the authors conduct an experimental investigation of the process of electrodialysis of reconstituted tobacco extract. The relevance of the study is dictated by the ecological safety of the use of electrodialysis to remove inorganic ions from tobacco extract. The authors provide measurements of the time dependencies of current, voltage, pH and conductivity for different concentrations of tobacco extract during electrodialysis. The experimental setup and the technique for measuring these parameters are described in detail.

Response. Thank you very much for your encouraging remarks. We greatly considered your expert comments and response to each query as follows;

  1. The abstract states "We conducted a detailed study to examine the inorganic components and their contributions to harmful ingredients in cigarette smoke". Authors should explain which harmful ingredients are involved and how they have been investigated.

Response. 1 In our previous study [Membr. Water Treat. 7(4) (2016) 341-353], we have already conducted detailed research about the removal of harmful products such as formaldehyde, CO, HCN, NH3, phenol, crotonaldehyde, and total particulate matter (TPM). The compounds were separated using ED, and their analysis was tested by sophisticated analytical equipment. The detailed procedure can be reached in the Journal of Thermal Analysis and Calorimetry volume 118, pages1747–1753 (2014). However, in the current study, we mainly concentrated on removing monovalent and divalent ions and various effects that could accelerate the desalination.

2 The introduction should explicitly describe the novelty of this article.

Response. 2 We revised the introduction section and added to the point statement, which testifies the importance and novelty of this statement. “The reconstituted tobacco extract with ED treatment showed improved performances on the taste and delivery of toxic mainstream tobacco smoke compared to that without the ED treatment” “The papermaking process reconstituted tobacco product that contains tobacco stems and fines as primary raw materials and is produced using papermaking technology. The principal procedure is as follows: (i) place tobacco raw material into the water to isolate soluble and insoluble; (ii) the insoluble will go via refining process and paper machine for fiber sheet while the soluble extracts will be sized back to the base sheet after concentrated. The sized sheet will be dried and ultimately assembled into PM recon” and “Herein, we developed readily adaptable methods to monitor the desalination of reconstituted tobacco extract and elucidated the evolutions of the ED process based on ion-selective membranes. Therefore, the main objectives of this research were to investigate the stability of the ED process for desalting reconstituted tobacco extract and enlighten the application of ED for industrial application”.

3 In Figure 1, will the red arrows indicate that the CEM is not permeable to Ca2+, Mg2+ ions? At the same time, the conclusion indicates « For tobacco extract of different concentrations, monovalent ions' removal ratios are as high as for divalent ions.» On the basis of what measurements did the authors arrive at this conclusion?

Response. 3 We are sorry for this mistake; we have redrawn the Figure. 1 and corrected the passage of ions across the cation exchange membrane. The bigger the hydrated size of the divalent cations, the lower mobility, and the membrane can efficiently fractionate the divalent ions and decrease their passage across the membrane, improving the removal ratio high for divalent ions.

4 The abbreviations AEC, CEC are not described on page 6.

Response. 4 Thank you for your suggestion; we revisited page 6 and provided the full description for the AEC and CEC., i.e., Anion exchange capacity (AEC) and Cation exchange capacity (CEC).

5 The description below figure 2 does not contain the letters "C", "D".

Response. 5. Thank you. We have redrawn all figures clearly and explained each caption with individual letters in each Fig(s) as suggested by Reviewer 2 (Q.2).

6 The meaning of the sentence "The sugar/nicotine ratio trend is consistent with total sugar." is not clear (page 9).

Response. 6.  We rephrased the sentence and added the following explanation for understanding “The ratio of sugar to nicotine would give a balance of opposing effects and thus serve as a good smoking quality indicator. A high percentage may indicate mildness and smoothness, while a meagre ratio may indicate harsh, irritating smoke. If the ratio is too high, it may mean that the tobacco is too mild to be pleasing to the smoker. High sugar content agreeing with nicotine level is the most desirable quality for smoking”. Please see page 9.

6 The signature of Figure 7 «in concentration chamber of desalination chamber» should be corrected.

Response. 7 Thank you for pointing out this mistake. We corrected “the variation curves of main ions in the concentrated and desalinated chambers at different pH values”.

Reviewer 2 Report

In the article "A comprehensive analysis of inorganic ions and their selective removal from the reconstituted tobacco extract using an electrodialysis" the authors conduct an experimental investigation of the process of electrodialysis of reconstituted tobacco extract. The relevance of the study is dictated by the ecological safety of the use of electrodialysis to remove inorganic ions from tobacco extract.

The authors provide measurements of the time dependencies of current, voltage, pH and conductivity for different concentrations of tobacco extract during electrodialysis. The experimental setup and the technique for measuring these parameters are described in detail.

Several questions/comments were raised when reading the article:

1) The abstract states "We conducted a detailed study to examine the inorganic components and their contributions to harmful ingredients in cigarette smoke". Authors should explain which harmful ingredients are involved and how they have been investigated.

2) The introduction should explicitly describe the novelty of this article.

3) In Figure 1, will the red arrows indicate that the CEM is not permeable to Ca2+, Mg2+ ions? At the same time, the conclusion indicates « For tobacco extract of different concentrations, monovalent ions' removal ratios are as high as for divalent ions.» On the basis of what measurements did the authors arrive at this conclusion?

4) The abbreviations AEC, CEC are not described on page 6.

5) The description below figure 2 does not contain the letters "C", "D".

6) The meaning of the sentence "The sugar/nicotine ratio trend is consistent with total sugar." is not clear (page 9).

7) The signature of Figure 7 «in concentration chamber of desalination chamber» should be corrected

Author Response

Reviewer 2

The paper 'A comprehensive analysis of inorganic ions and their selective removal from the reconstituted tobacco extract using an electrodialysis' by Noor et al. summarizes the reconstituted tobacco extract findings. However, it is not just a 'paper about papers' but delivers further analysis and endeavors to extract the most noteworthy results. It is well written and well balanced between industrial and academic collaboration. It deserves to be published and is a practical contribution to this journal. In order to make it more readily accessible, several questions and comments were raised when reading the article:

Response. Thank you very much for your encouraging remarks. We greatly considered your expert comments and response to each query as follows;

1 It is suggested to redraft the abstract section and make it more conclusive.

Response. 1 We revised the abstract section; please see the revised manuscript abstract.

2 The figures' quality is inferior, and it is hard for the reader to extract a finding.

Response. 2 We have redrawn all figures clearly and explained each caption with individual letters in each Fig(s).

3 Please provide more detail about the title of each figure, and cite it in the main text.

Response. 3 We have redrawn all figures clearly and added more detail about each figure with individual letters.

  1. 4 Section 3.1 and 3.4 needs to be combined/discussed in one section.

Response. 4. We agree with the reviewer that it should be combined in one section. However, the test conditions are not the same; therefore, discussing them in separate sections is more explicit.

5 How has the concentration of ions been calculated? Please provide a mathematical equation.

Response. 5. We have provided the concentration in mg/L directly from the ICP machine. We did not use the equation in the present work to calculate the ion flux. For ion flux, please refer to our previous work. Ref.  J Polym Sci.2021;1-13, J. Membr. Sci., 639, (2021), 119757, and Sep. Purif. Technol., 254 (2021) 117619.

6 Please add some recent examples and references for ion-selective ion exchange membranes and ED applications, such as Journal of Membrane Science, 2021, 634, 119407; Desalination 2022, 531,115690, etc.

Response. 6. Thank you very much for your suggestion. We have added a few references to show the recent advances in the ion-selective membrane and the application of the ED process.

Reviewer 3 Report

Reviewer Comments

I recommend this manuscript to publish in Membranes after these minor improvements:

1.      The abstract is too long. It should be re-write

2.      The introduction of the manuscript should be improved by reporting recent work. There are given references of old studies.

3.      In line Figure 5, why the result are not provided on neutral pH (pH= 7)?

4.      In Figure 7, why the result are not provided on neutral pH (pH= 7)?

5.      Cites these articles.

(i)                 doi:10.3390/ma9050365.

(ii)               Desalination 402 (2017) 10–18.

6.      Check spelling and other language errors throughout.

7.      The English need to improve throughout the manuscript.  

Author Response

Reviewer 3

I recommend this manuscript to publish in Membranes after these minor improvements;

Response; Thank you very much for your positive comments.

1 The abstract is too long. It should be re-written.

Response. 1 Than you for your comment. We have revised the abstract.

2 The introduction of the manuscript should be improved by reporting recent work. There are given references of old studies.

Response. 2, Yes, you are right; we have changed and provided recent references.

3 In Figure 5, why the result is not provided on neutral pH (pH= 7)?

Response. 3 Yes, you are right and should be tested. However, we focused only on low and high pH, i.e., a higher pH may result in the precipitation of divalent ions, lowering their availability. In contrast, at very low pH, the excess H+ can carry the charges across the membrane, impeding the transport of other ions.

4 In Figure 7, why the result is not provided on neutral pH (pH= 7)?

Response. 5 Yes, you are right and should be tested. However, we focused only on low and high pH, i.e., a higher pH may result in the precipitation of divalent ions, lowering their availability. In contrast, at very low pH, the excess H+ can carry the charges across the membrane, impeding the transport of other ions.

  1. 5 Cites these articles.
  2. doi:10.3390/ma9050365.
  3. Desalination 402 (2017) 10–18.

Response. 5. Thank you. The current assignment does not harmonize with the references provided by the honorable reviewer. However, we believe it would be very knowledgeable for our future job associated with the electrodialysis process.

6 Check spelling and other language errors throughout.

Response. 6. We have revised the manuscript for grammar mistakes.

  1. 7 The English need to improve throughout the manuscript.

Response. 7. Yes, you are right; we have checked the whole manuscript for English correction to make it best for the academic point of view.

Round 2

Reviewer 2 Report

All right!